# Development of Synthesis and Application of High Molecular Weight Poly(Methyl Methacrylate)

**DOI:** 10.3390/polym14132632

**Published:** 2022-06-28

**Authors:** Ming Yuan, Dayun Huang, Yixuan Zhao

**Affiliations:** Department of Chemistry and Chemical Engineering, College of Ecology, Lishui University, Lishui 323000, China; dayunhuang@lsu.edu.cn (D.H.); zyx13566196913@163.com (Y.Z.)

**Keywords:** poly(methyl methacrylate), high molecular weight, polymerization, application

## Abstract

Poly(methyl methacrylate) (PMMA) is widely used in aviation, architecture, medical treatment, optical instruments and other fields because of its good transparency, chemical stability and electrical insulation. However, the application of PMMA largely depends on its physical properties. Mechanical properties such as tensile strength, fracture surface energy, shear modulus and Young’s modulus are increased with the increase in molecular weight. Consequently, it is of great significance to synthesize high molecular weight PMMA. In this article, we review the application of conventional free radical polymerization, atom transfer radical polymerization (ATRP) and coordination polymerization for preparing high molecular weight PMMA. The mechanisms of these polymerizations are discussed. In addition, applications of PMMA are also summarized.

## 1. Introduction

Poly(methyl methacrylate) (PMMA), commonly known as plexiglass, is a kind of polymer synthesized by free radical polymerization, ionic polymerization and coordination polymerization [1,2,3]. Compared with ordinary glass (silica glass), PMMA not only has a relatively low density, but also has a better crushing resistance. PMMA can be used as an excellent functional polymer material due to its good biocompatibility [4]. Moreover, PMMA is also widely used in medical technologies and photoelectronic devices as well as other areas [5].

In general, the PMMA with a number-average molecular weight exceeding 10^6^ Da can be defined as ultrahigh molecular weight PMMA and the PMMA with a number-average molecular weight exceeding 10^5^ Da can be defined as high molecular weight PMMA [6,7]. Low molecular weight PMMA has limited applications because of its poor heat resistance, poor wear resistance, poor organic solvent resistance, low hardness and easy combustion features. It has long been known that the mechanical properties of PMMA largely depend on its molecular weight [8,9,10,11,12]. Consequently, it is of great importance to develop feasible polymerization routes for the synthesis of high molecular weight PMMA. Conventional free radical polymerizations including emulsion polymerization and suspension polymerization can be used to synthesize high molecular weight PMMA [13]. Since it was discovered by Matyjaszewski in 1995 [14], atom transfer radical polymerization (ATRP) has become a successful approach for the preparation of well-defined polymers. The synthesis of high molecular PMMA by ATRP can also be realized. Other synthesis routes such as coordination polymerization, reversible addition-fragmentation chain transfer radical polymerization (RAFT) and plasma-initiated polymerization are also employed to prepare high molecular weight PMMA [15].

In this review, synthesizing high molecular weight PMMA by conventional radical polymerization, atom transfer radical polymerization and coordination polymerization from the aspects of catalytic activity, initiation activity, molecular weight and mechanism are discussed. Other polymerization methods, including reversible addition-fragmentation chain transfer radical polymerization (RAFT) and plasma-initiated polymerization, are also described. Applications of high molecular weight PMMA and its copolymers on medical, electricity and other areas are summarized. In addition, we briefly introduce some achievements of our group in the synthesis of ultra-high molecular weight vinyl polymers.

## 2. Synthesis of High Molecular Weight PMMA

### 2.1. Free Radical Polymerization

Free radical polymerization is one of the most important technologies for industrial production of polymers, owing to its simple operation and good reproducibility property. Free radical polymerization including conventional radical polymerization, ATRP and RAFT are effective methods for preparing high molecular weight PMMA. In this part, applications of conventional radical polymerization and ATRP for synthesizing high molecular weight PMMA are mainly discussed.

#### 2.1.1. Conventional Radical Polymerization

It is difficult to obtain the polymer with controlled molecular weight and narrow molecular weight distribution by conventional radical polymerization because of the slow initiation, fast growth and fast termination characteristics of conventional free radical polymerization. However, conventional free radical polymerization has an excellent advantage for preparing high molecular weight polymers.

In general, the rate of free radical polymerization of vinyl monomer and the molecular weight of its vinyl polymer mainly depend on initiation, propagation, and termination steps. In a free radical polymerization, the rate of polymerization (*R*_p_) should follow the equation of *R*_p_ = *k*_p_[M][I]^0.5^(*fk*_d_/*k*_t_)^0.5^ (*k*_p_ is the propagation rate constant; *k*_d_ is the decomposition rate constant; *k*_t_ represents the termination rate constant; *f* represents the initiator efficiency) and the kinetic chain length (υ) should follow the equation of υ = *k*_p_[M]/2(*fk*_d_*k*_t_[I])^1/2^ [16]. It indicates that the molecular weight of the polymer is increased with the decrease in initiator concentration. Therefore, a low initiator concentration is needed for preparing high molecular weight polymers. However, a low initiator concentration will cause the decrease in polymerization rate (*R*_p_) and it needs much longer reaction time to realize the preparation of high molecular weight polymer with a high yield. Moreover, the polymerization will not occur under an extremely low initiator concentration [16]. Therefore, a suitable reaction system is essential for the synthesis of high molecular weight polymers.

Azodiisobutyronitrile (AIBN) and benzoyl peroxide (BPO) are the most commonly used initiators in conventional radical polymerization. Figure 1 shows the mechanism of conventional free radical polymerization of MMA initiated by BPO.

In a conventional free radical polymerization of MMA, PMMA can be synthesized by bulk polymerization, emulsion polymerization and suspension polymerization. High molecular weight PMMA can be prepared by bulk polymerization. However, it is difficult to remove the generated heat in the polymerization, resulting in a generation of gel effect and side reactions. Therefore, there is still obstacles to preparing high molecular weight PMMA with high yield through bulk polymerization. Compared with ionic polymerization, conventional free radical polymerization is more resistant to water. Consequently, dispersion polymerization, suspension polymerization and emulsion polymerization using water or other solvents as medium are proposed and used to synthesize high molecular weight PMMA.

Commonly, in a free radical polymerization, the initiating and chain growing free radicals are easily quenched by oxygen (O_2_), generating stable and inactive peroxyl radicals. Therefore, it is important to solve the problem of oxygen inhibition in a radical polymerization. Degassing of the polymerization environment with an inert inactive gas, such as argon or nitrogen, is an effective method to prevent oxygen inhibition.

Hsiao et al. [17] report a polymerization of MMA by using poly(1,1-dihydroperfluorooctyl acrylate) [poly(FOA)] as a stabilizer, AIBN as an initiator, and supercritical CO_2_ as a reaction medium. Due to the special properties of supercritical CO_2_, the dissolution capacity of the solution is easily controlled by changing temperature and pressure. They find that the molecular weight of PMMA is influenced by stabilizer concentration and monomer concentration. A relatively higher monomer concentration is beneficial for preparing high molecular weight PMMA, and the PMMA with a number-average molecular weight (*M*_n_) of 3.16 × 10^5^ Da has been synthesized. Subsequently, Hsiao et al. [18] also report a dispersion polymerization of MMA with poly(FOA) as a stabilizer, AIBN as an initiator, and supercritical CO_2_ as a reaction medium in the presence of helium. The high molecular weight PMMA with a *M*_n_ of 3.65 × 10^5^ Da is synthesized. The solvency of supercritical CO_2_ has an important influence on the particle size and particle size distribution of the synthesized PMMA. The solvency of supercritical CO_2_ can be adjusted by the addition of helium. Consequently, the use of helium offers an attractive method for controlling the particle size and size distribution.

Lepilleur et al. [19] report a dispersion polymerization of MMA with poly(methyl methacrylate-*co*-hydroxyethyl methacrylate)-*g*-poly(perfluoropropylene oxide) [P(MMA-*co*-HEMA)-*g*-PFPO] as a stabilizer, AIBN as an initiator and supercritical CO_2_ as a reaction medium. The effect of P(MMA-*co*-HEMA)-*g*-PFPO hyperbranched polymers with different molecular weights on molecular weight of the synthesized PMMA is investigated, and the PMMA with a high molecular weight of 3.55 × 10^5^ Da has been synthesized. Actually, the solubility of P(MMA-*co*-HEMA)-*g*-PFPO hyperbranched polymers has a significant effect on the molecular weight distribution of PMMA, and P(MMA-*co*-HEMA)-*g*-PFPO with good solubility is beneficial for synthesizing narrow molecular weight distribution PMMA.

Although the addition of stabilizers improves the dispersing ability of the monomer in the polymerization system, it brings increased costs of production and post-processing. Wang and coworkers find that the dispersion polymerization of MMA using AIBN as an initiator and supercritical CO_2_ as a reaction medium in the absence of stabilizers can realize the synthesis of high molecular weight PMMA [7]. It seems that the vapor-liquid equilibrium (VLE) is one of the key factors of the molecular weight. For instance, the polymerization conducted under 28.5–11.8 MPa, the PMMA with a low molecular weight is obtained and MMA-CO_2_ mixture is in a homogeneous phase. However, when the polymerization conducted below 9.2 MPa, the MMA-CO_2_ mixture is at VLE and the high molecular weight PMMA (*M*_n_ = 1.31 × 10^5^ Da) is synthesized.

Similarly to supercritical CO_2_, ionic liquids (ILs) are also an environmentally friendly reaction medium. IL is a kind of low melting point (<100 °C) salt composed of a large volume of organic cation and an organic or inorganic anion [20,21]. ILs are widely investigated because of their advantages such as being non-volatile and non-flammable, and having high solubility and recyclability. ILs can be used as plasticizers for PMMA and poly(vinylidene fluoride) [22,23,24], and can also affect the formation of functional polymer materials such as coordination polymers [25], silicon-containing gels [26], and macroporous membranes [27].

Vygodskii et al. [28] report a free radical polymerization of MMA with AIBN as an initiator and ILs as a solvent, and ultrahigh molecular weight PMMA with a weight-average molecular weight (*M*_w_) of 5.77 × 10^6^ Da is synthesized. The used ILs are mainly composed of 1,3-dialkylimidazole cations and ${\mathrm{BF}}_{4}^{-}$ ${\mathrm{PF}}_{6}^{-}$, ${\mathrm{SbF}}_{6}^{-}$, CF_3_${\mathrm{SO}}_{3}^{-}$, (CF_3_SO_3_)_2_$N^{-}$, CH_3_${\mathrm{COO}}^{-}$, (CF_3_CF_2_)_3_${\mathrm{PF}}_{3}^{-}$ and other anions. They find that the molecular weight is affected by ILs, and ILs with a long carbon chain length are not beneficial forpreparing high molecular weight PMMA.

Suspension polymerization has a similar polymerization mechanism for bulk polymerization. However, it is easier to control the heat and the viscosity in suspension polymerization. Therefore, suspension polymerization is more promising for synthesizing high molecular weight PMMA with high monomer conversion. Moreover, suspension polymerization is also applied to prepare polymer-inorganic nanocomposites, which is widely used in optics [29], photoconductors [30,31], and electronic equipment areas [32]. Regrettably, the polymer produced by suspension polymerization inevitably contains a small amount of dispersant residue, which will affect the electrical properties of the product.

In fact, alongside a high molar ratio of monomer to initiator, a relatively low polymerization temperature is also needed for preparing high molecular weight polymers using suspension polymerization. This is because of the suppression of chain termination reaction and chain transfer reaction under low polymerization temperature. Therefore, a low-temperature initiator such as 2,2′-azobis (2,4-dimethylvaleronitrile) (ADMVN) can use to initiate the suspension polymerization of MMA at a low temperature.

Lyoo et al. [16] report a suspension polymerization of MMA conducted at 25 °C with poly (vinyl alcohol) (PVA) as a suspending agent and 2,2′-azobis (2,4-dimethylvaleronitrile) (ADMVN) as an initiator. The ultrahigh molecular weight PMMA with a *M*_n_ of 3.61 × 10^6^ Da and monomer conversion of 83% is synthesized at 25 °C with a polymerization time of 96 h. The feed ratio has an observable effect on polymerization and the molecular weight of PMMA is increased with the decrease in the molar ratio of initiator to monomer. However, the polymerization fails to be realized when the molar ratio of initiator to monomer decreased to 5 × 10^−5^. In the presence of silver nanoparticles, ultrahigh molecular weight (*M*_υ_ = 3.70 × 10^6^ Da; *M*_υ_ is viscosity-average molecular weight) PMMA is also synthesized by suspension polymerization with PVA as a suspending agent and ADMVN as an initiator [33]. Actually, the advantage of addition of silver microspheres not only promotes the formation of PMMA/silver microspheres but is also beneficial in improving the polymerization rate.

Emulsion polymerization is one of the most commonly used methods to synthesize high molecular weight polymer owing to its high reactivity and property of simultaneous increase in the polymerization rate and molecular weight. However, it is unavoidable to add a large amount of emulsifier in emulsion polymerization, resulting in a decrement in the properties of synthesized polymers. Consequently, both differential microemulsion polymerization and soap free emulsion polymerization are developed and applied to the synthesis of high molecular weight polymers.

Jiang et al. [34]. propose a microemulsion polymerization of MMA using sodium dodecylsulfate (SDS) as a surfactant, 1-pentanol (n-Pt) as a co-surfactant, and ammonium persulfate (APS) as an initiator. The dosage of surfactant is obviously decreased, which provides beneficial conditions for the interaction between the free radicals and polymer particles. Therefore, it is easier to synthesize PMMA with narrow molecular weight distribution and high molecular weight. By optimizing the conditions, the high molecular weight PMMA with a weight-average molecular weight of 7.60 × 10^5^ Da and a molecular weight distribution (*M*_w_/*M*_n_, PDI) of 1.40 has been successfully synthesized in their group. Moreover, this polymerization method provides a new route for preparing multi-chain PMMA particles. Norakankorn et al. [35] report a differential microemulsion polymerization of MMA with SDS as surfactant and AIBN as initiator, and PMMA with a high number-average molecular weight of 1.00 × 10^6^ Da has been successfully prepared under a condition of [MMA]:[SDS] = 130:1 (weight ratio). As an oil-soluble initiator is used in this reaction, the particle nucleation is believed to take place in micelles and follow the heterogeneous nucleation mechanism.

In recent years, the semiconductor nanoparticles composed of transition metal ions and group VI and group VII ions have attracted much attention of chemists due to their wide applications in solar cells, photovoltaics, biosensors, and environmental protection areas [36,37]. Based on this, the semiconductor nanoparticles mediated photopolymerization of vinyl monomers has attracted the great attention of chemists [38,39]. In addition, the use of O_2_ in the initiation mechanism has also been investigated in photopolymerization [40,41].

In the presence of O_2_, Dadashi-Silab et al. [37] find that the ZnO or Fe/ZnO semiconductor nanoparticles can be used as photoinitiators for free radical polymerization of MMA under the irradiation of UV light. The addition of triethylamine (TEA) and diphenyliodonium hexafluorophosphate (Ph_2_I^+^PF_6_^−^) can promote the polymerization, and the PMMA with a *M*_n_ of 2.10 × 10^5^ Da is synthesized via Fe/ZnO/TEA system. They believe that the addition of TEA and Ph_2_I^+^PF_6_^−^ will promote the formation of radical capable of initiating the polymerization. Mandal et al. [42] report a photopolymerization of MMA with ZnO or ZnO/Ag nanoparticles as a catalyst under the irradiation of UV light. A relatively high polymerization rate is achieved in the ZnO/Ag catalyst system, and the monomer conversion is reached to 82% after 9 h. In the ZnO catalyst system, the polymerization rate is comparatively slower, and the monomer conversion is only upped to 50% after 10 h. The molecular weight of PMMA produced in the ZnO/Ag system is higher than in the ZnO system, and the PMMA with a *M*_n_ of 1.94 × 10^5^ Da is prepared. The mechanism has also been investigated, finding that the hydroxyl radical generated due to the photolysis of water by ZnO/Ag nanoparticle is the active species for the polymerization.

#### 2.1.2. Atom Transfer Radical Polymerization (ATRP)

The high molecular weight polymer with a controlled structure is widely used in crystallization characteristics control, rheology modification, mechanical performance and morphology development [43]. Conventional free radical polymerization has the advantage of preparing high molecular weight polymers. However, it is difficult to prepare polymers with a controlled structure and a narrow molecular weight distribution through conventional free radical polymerization. In 1995, Matyjaszewski and coworkers firstly report a “living”/controlled polymerization of styrene via atom transfer radical polymerization (ATRP) [14]. In 2001, Matyjaszewski and coworkers systematically summarized the development of ATRP [44]. In this period, a series of catalytic systems were developed and then used for the preparation of well-defined polymers. Nowadays, ATRP also becomes an effective route for preparing high molecular weight polymers [45,46]. The mechanism of copper mediated normal ATRP of MMA is shown in Figure 2.

As shown in Figure 2, under the catalysis of a transition metal complex (Cu^I^X/L), the radical (R•) is formed via a reversible redox process which undergoes an abstraction of a halogen atom (X) from a dormant species (R-X). The chain growth reaction is realized via the addition of monomer to the active species (R• or RM_n_•). There exists reversible dynamic equilibrium (*k*_a_/*k*_d_) between active specie (RM_n_•) and dormant specie (RM_n_X). Owing to this, the molecular weight of the polymer is increased with the increase in monomer conversion. Actually, similarly to conventional free radical polymerization, termination reactions such as radical coupling and disproportionation also occur in ATRP. Differently, there is no more than a few percent of the polymer chains which undergo termination in a well-controlled ATRP [44]. Oxidized metal complexes (Cu^II^X_2_/L) generated in the system have a significant effect on the chain termination during the initial stage of the polymerization. The formation of Cu^II^X_2_/L will reduce the chain growth radical concentration and thereby decrease the occurrence of termination reaction.

Typically, the normal ATRP system is composed of monomer, organic halide initiator and catalyst. A variety of monomers such as styrenes and (meth)acrylates are successfully polymerized via ATRP. Each monomer has its own suitable atom transfer equilibrium constant (*K*_eq_ = *k*_a_/*k*_d_) for its active and dormant species. ATRP will occur very slowly or even fail if the *K*_eq_ is too small. Conversely, a large *K*_eq_ will cause a high radical concentration and thereby cause a large amount of termination [45]. Therefore, it is important to control the active species concentration and radical deactivation rate in the polymerization system.

The initiator is also a key factor in ATRP owing to its control of the number of macromolecular growth chain. Using a good ATRP initiator, the initiation should be fast and quantitative. Therefore, the number of macromolecular growth chain can be controlled. In addition, the selection of initiator will affect the formation of polymer. One of the reasons is that the migration speed of the halide group (X) will affect the PDI of synthesized polymer and a fast migration speed of the halide group between the chain growth radical and the transition-metal complex is beneficial to obtain a polymer with small PDI. It means that a successful ATRP should have features such as a small amount of terminated chains and a uniform growth of all the chains.

In order to prepare high molecular weight PMMA, we should firstly know the kinetics of ATRP. Typically, in a copper mediated normal ATRP, the polymerization rate (*R*_p_) follows the equation of *R*_p_ = *k*_p_[M][P*] = *k*_p_*K*_eq_[M][I_0_] × [Cu^I^]/[Cu^II^] and the degree of polymerization (DP) follows the equation of DP = ∆M/[I_0_] [44,47]. Therefore, we can know that a high molar ratio of monomer to initiator is necessary for preparing high molecular weight polymers. In addition, the decrease in initiator concentration can also decrease the occurrence of termination reaction [43]. However, this will inevitably cause a decrease in polymerization rate.

Agarwal et al. [43] report a ATRP of MMA with phenyl 2-bromo-2-methylpropionate (BMPE) as an initiator, CuCl as a catalyst and *N*,*N*,*N*′,*N*″,*N*″-pentamethyldiethylene triamine (PMDETA) or 4,4′-dinonyl-2,2′-bipyridine (dnNbpy) as a ligand. The molecular weight synthesized in PMDETA system is deviated from the theoretical values, which may be caused by coupling termination of free radicals. High molecular weight PMMA with a *M*_n_ of 3.67 × 10^5^ Da and molecular weight distribution of 1.20 is prepared in dnNbpy system with a molar ratio of [MMA]:[BMPE]:[CuCl] = 6400:1:6.

The star polymer attracts a lot of interests because of its branched structure. Similarly to the preparation of linear polymers, the preparation of high molecular weight star polymers also requires a low extent of chain transfer and chain termination, high initiation efficiency, and moderate polymerization rate. Agarwal and coworkers [48] report a ATRP of MMA using 2,3,6,7,10,11-triphenylhexyl-2-bromo-2-methylpropionate (TP6Br) or ethylene glycol bis[3,4,5-tris(2-Bromo-2-methylpropionate)]benzoate (EG(Bz3Br)_2_) as an initiator, CuCl as a catalyst, and dnNbpy as a ligand. The high molecular weight six-arm star PMMA with a *M*_w_ of 2.35 × 10^6^ Da is synthesized using TP6Br as initiator. In addition, a six-arm star PMMA with a *M*_w_ of 2.10 × 10^6^ Da is successfully synthesized using EG(Bz3Br)_2_ as initiator.

A high concentration of catalyst is often required to ensure the controllability of normal ATRP, which will cause an increment in side reaction and the decrement in molecular weight. The concentration of the catalyst can be significantly decreased in an electron transfer activation regeneration catalyst atomic transfer radical polymerization (ARGET ATRP), resulting in a decrease in side reactions. Therefore, the ARGET ATRP may be applied to prepare higher molecular weight polymers. The mechanism of copper mediated ARGET ATRP of MMA is shown in Figure 3.

Different from normal ATRP, in a copper mediated ARGET ATRP, the transition metal complex in its higher oxidation state is added to the system. The activator species (Cu^I^) are constantly regenerated from the transition metal complex in its higher oxidation state (Cu^II^) by the reduction in reducing agent. Therefore, the amount of Cu(II) species is much lower than the amount of reducing agent. Actually, the use of Cu(II) species can be reduced to around 10 ppm. The extent of the side reaction is reduced owing to the drastically smaller amounts of required Cu(II) species and thereby high molecular weight polymer can be prepared via ARGET ATRP [50].

Alkyl dithioester is a commonly used RAFT chain transfer agent (CTA), which can be activated by free radicals or ATRP catalysts to initiate the polymerization of MMA. Therefore, CTA acted as an alkyl pseudohalide is also an ATRP initiator. Matyjaszewski et al. [50] report an ARGET ATRP of MMA by using cumyl dithiobenzoate (CDB) as an initiator, CuBr as a catalyst, copper powder or copper wire as a reducing agent. The effects of catalyst concentration on the polymerization rate and the molecular weight of PMMA are investigated. Within a certain catalyst concentration range (0.33–50 ppm), increasing the catalyst concentration will cause an increment in the polymerization rate and a decrement in the molecular weight. The high molecular weight PMMA (*M*_n_ = 1.25 × 10^6^ Da) with a narrow molecular weight distribution (PDI = 1.21) is synthesized. Moreover, the high molecular weight (*M*_n_ = 2.22 × 10^5^ Da) poly(styrene) (PS) and high molecular weight (*M*_n_ = 1.80 × 10^6^ Da) poly(methyl methacrylate)-*b*-poly(butyl methacrylate) (PMMA-*b*-PBMA) are also synthesized. Matyjaszewski et al. [51] also report a surface-initiated ATRP (SI-ATRP) of MMA using SiO_2_-Br as an initiator. The polymerization of MMA can be successfully realized with a low concentration of catalysts (12.5 ppm) and producing a high molecular weight (*M*_n_ = 1.70 × 10^6^ Da) SiO_2_-*b*-PMMA.

Youk et al. [52] report that the high molecular weight 3-arm PMMA can be synthesized via ARGET ATRP by using 1,3,5-tris(2-bromoisobutoxy)benzene (TBIB) as an initiator, CuBr_2_ as a catalyst, PMDETA as a ligand and tin(II) 2-ethylhexanoate [Sn(EH)_2_] as a reducing agent. The effects of initiator concentration, catalyst concentration and reducing agent concentration on the molecular weight of PMMA have been investigated, and the high molecular weight PMMA (*M*_n_ = 5.70 × 10^5^ Da, PDI = 1.36) is successfully prepared under a molar ratio of [CuBr_2_]:[PMDETA]:[Sn(EH)_2_] = 0.1:1.5:1.5.

In a free radical polymerization system, the molecular weight of synthesized polymers is proportional to the ratio of *k*_p_^2^/*k*_t_, and the propagation rate constant (*k*_p_) can be improved by increasing the pressure [53,54]. Therefore, increasing polymerization pressure is beneficial to prepare high molecular weight polymer.

Matyjaszewski et al. [55] report a ATRP of MMA by using CuBr_2_ as a catalyst, EBiB as an initiator, tris(2-pyridylmethyl)amine (TPMA) as a ligand and ascorbic acid (AsAc) as a reducing agent. The effect of pressure (10^−3^, 3, 6, and 10 kbar) on the polymerization is investigated. The polymerization proceeded at 3 kbar has a relatively higher polymerization rate than the polymerization conducted at atmospheric pressure. The high molecular weight (*M*_n_ = 5.93 × 10^5^ Da) PMMA can be synthesized at 10 kbar, but a relatively low monomer conversion (46%) and high molecular weight distribution (PDI = 1.96) are obtained. The polymerization rate and molecular weight can both be increased at 6 kbar, giving a well-defined PMMA with a *M*_n_ of 1.88 × 10^6^ Da.

In the presence of a low catalyst concentration ([CuCl] = 14 mM), Fukuda et al. [53] systematically investigated the effect of pressure (0.1-500 MPa) on the polymerization of MMA. The controllability of the polymerization is increased by increasing the pressure, producing PMMA with a narrow molecular weight distribution (PDI = 1.20–1.60). Under a 500 MPa pressure condition, the polymerization rate is 20 times faster than that at ambient pressure and produces the PMMA with a *M*_n_ of 5.00 × 10^5^ Da and PDI of 1.25. However, the controllability of the system will decrease at higher monomer conversion. In order to improve the controllability, 0.03 mM CuCl_2_ used as a deactivator is added into the original feed. The polymerization is also conducted at 500 MPa, finding that the polymerization rate and PDI are both decreased, and giving a well-defined high molecular weight PMMA (*M*_n_ = 1.50 × 10^6^ Da, PDI = 1.25). The effect of initiator concentration on the molecular weight is also investigated, and the PMMA with a high molecular weight (*M*_n_ = 3.60 × 10^6^ Da) is synthesized with a low initiator concentration of 0.047 mM.

### 2.2. Coordination Polymerization

Coordination polymerization is also an effective method for preparing high molecular weight PMMA [56]. The mechanism of coordination polymerization of MMA is shown in Figure 4.

In a coordination polymerization, the catalytic activity and the polymer molecular weight are significantly affected by metal, ligand, temperature, cocatalyst and molar ratio of monomer to catalyst [57,58,59,60,61,62,63,64,65]. Generally, increasing the molar ratio of monomer/catalyst will cause an increment in the molecular weight. The steric encumbrance in the ligand around the metal center has an obvious influence on the catalytic activity. For instance, for a pyrazole-based ligand, if the pyrazole moiety has a substituent, the catalytic activity will be enhanced in the polymerization of MMA. The bulkier the substituents on pyrazole moiety, the higher the catalytic activity for MMA polymerization. The solubility of the catalyst will also affect the catalytic activity. For a typical Ziegler-Natta catalyst such as TiCl_4_-AlR_3_, if the halogen has been substituted by cyclopentadienyl (Cp), the obtained catalyst (Cp_2_TiCl_2_-AlR_3_) will be soluble in the organic solvent. The polymerization rate can be increased by increasing the temperature. However, this will cause a decrement in the molecular weight. According to the literature [57,58,59,60,61,62,63,64,65,66,67,68,69], temperature range from 30 °C to 80 °C might be conducive to the synthesis of high molecular weight PMMA in a coordination polymerization. The cocatalyst (AlR_3_) is also a key factor for the polymerization of MMA. Within a certain range, both polymerization rate and polymer molecular weight can be increased by increasing the concentration of cocatalyst. However, a high concentration of cocatalyst will cause a decrement in molecular weight, owing to the chain transfer reaction.

The electron-donating atoms such as O and N in polar monomers are easy to coordinate with the catalyst, forming a stable complex and destroying the catalytic ability. Consequently, the typical Ziegler-Natta catalyst is not quite suitable for the polymerization of polar monomers such as AN and MMA. Homogeneous metallocene activated by aluminum alkyl (AlR_3_) can induce a coordination polymerization of MMA but showing a relatively poorer efficiency [57]. Lanthanocene catalysts have high activity in the coordination polymerization of MMA, and easily produce high molecular weight PMMA [56].

Sun et al. [57] report a coordination polymerization of MMA by using ethylene bridged of titanocene chloride and samarocene (Sm-Ti) as a single component catalyst or using the triisobutylaluminum (TIBA) activated Sm-Ti as a catalyst. Both the binary catalyst and the single component catalyst are soluble in the polymerization system. The binary catalyst shows a higher activity than single component catalyst but provides PMMA with a relatively lower molecular weight. In the single component catalytic system, the molecular weight of PMMA is increased with the increase in temperature. However, the molecular weight of PMMA synthesized in binary catalytic system is decreased with the increase in temperature. By optimizing the reaction conditions, the PMMA with a high *M*_υ_ of 1.50 × 10^6^ Da and monomer conversion of 87% has been synthesized with a polymerization time of 20 h and temperature of 80 °C.

Schiff base is an important nitrogen donor ligand, and a large number of schiff bases are investigated due to their activity to reversibly bind oxygen [58]. Yousaf et al. [59] propose that lanthanocene complexes with schiff base derivatives or methoxyethylindenyl derivatives as ligands can be used as an effective catalyst for the polymerization of MMA. The catalyst is easily dissolved in toluene and thus the homogeneous catalysis can be realized. The effects of the catalyst such as [(CP)(Cl)Ln Schiff base (THF)] (Ln = Sm, Dy, Er, Y) and (COT)Ln (methoxyethylindenyl) (THF) (Ln = La, Nd, Sm, Dy, Er) on the polymerization are investigated. Compared to (COT)Ln (methoxyethylindenyl) (THF), the [(CP)(Cl)Ln Schiff base (THF)] shows a better activity in the polymerization. The effects of temperature and feed ratio on the polymerization have also been investigated, and the high molecular weight PMMA with a *M*_υ_ of 4.16 × 10^5^ Da has been synthesized with a polymerization time of 20 h and temperature of 70 °C.

Compared to the early transition metal, the later transition metal has a lower oxygen affinity and higher functional group tolerance. Therefore, the later transition metal can also be applied to the polymerization of polar monomers. For instance, late transition metals such as Cu(II), Pd(II), Zn(II), Cd(II) and Co(II) complexed with *N*-substituted pyridylamine ligands are used as the catalysts in the polymerization of MMA [60,61]. Lee and coworkers report that Zn(II) complexes bearing 2-iminomethylquinoline and 2-iminomethylpyridine based ligands can be applied to the polymerization of MMA [62]. The bidentate [(*NN*′)ZnCl_2_] complexed with modified methylaluminoxane (MMAO) is used as a catalyst and toluene is used as a solvent in the polymerization of MMA. The PMMA with a *M*_w_ of 9.62 × 10^5^ Da is prepared at 60 °C with a polymerization time of 2 h.

Lavastre et al. [63] report that CuCl_2_ or Cu(OAc)_2_ combined with *N*-tripod ligand is an effective catalyst for the polymerization of MMA in the presence of methylaluminoxane (MAO). The commonly used *N*-tripod ligands are shown in Figure 5.

CuCl_2_ or Cu(OAc)_2_ alone does not induce the polymerization, indicating that the *N*-tripod ligand is also a key factor for the polymerization. In Cu(OAc)_2_ catalyst systems, the PMMA is prepared with a yield of 23% to 40%. CuCl_2_ combined with ligands **1b** or **1g** gave a PMMA with a yield of 60%. The molecular weight of PMMA prepared in Cu(OAc)_2_ system is higher than that produced in CuCl_2_ system. In the presence of toluene, the PMMA with a *M*_n_ of 3.39 × 10^5^ Da is synthesized in Cu(OAc)_2_/**1b** system with a polymerization time of 4 h and temperature of 30 °C.

Lee and coworkers [64] report a coordination polymerization of MMA using the cobalt (II) complex combined with *N*,*N*-bis{(1-pyrazolyl)methyl}aniline (bpmaL1) ligand as a catalyst in the presence of MMAO. A series of bidentate [*N*,*N*]-cobalt (II) dichloride complexes are synthesized and presented in Figure 6.

Effects of catalysts on the polymerization rate and molecular weight have been studied. The 4w/MMAO (1.14 × 10^6^ g/mol-Co·h) shows the highest activity, 2w/MMAO (8.00 × 10^5^ g/mol-Co·h) and 3w/MMAO (8.73 × 10^5^ g/mol-Co·h) have a moderate activity, and 1w/MMAO (6.17 × 10^4^ g/mol-Co·h) presents the lowest activity. However, the 1w/MMAO system is more suitable for preparing high molecular weight PMMA. With a polymerization temperature of 60 °C, the PMMA with a *M*_n_ of 1.13 × 10^6^ Da and PDI of 1.75 has been synthesized in 1w/MMAO system. Later, Lee et al. [65] also report a polymerization of MMA by using 4-coordinate cobalt(II) complexes bearing *N*′-substituted *N*,*N*′,*N*-bis((1H-pyrazol-1-yl)methyl)amine derivatives ligands as catalysts in the presence of MMAO. In the presence of toluene, the high molecular weight PMMA with a *M*_w_ of 1.05 × 10^6^ Da is prepared at 60 °C with a polymerization time of 2 h.

Half-sandwich nickel(II) complexes [Ni(ŋ^5^-C_5_H_4_R)(X)(NHC) (R = H or alkyl, X = Cl, Br, I)] (NHC = N-heterocyclic carbenes) are an efficient catalyst for the polymerization of styrene and phenylacetylene [66,67]. The Ni(II) complexes such as Ni(acac)_2_ can be used as a catalyst for the polymerization of MMA in the presence of MAO [68]. In the presence of MAO, Buchowicz and coworkers find that the [Ni(ŋ^5^-C_5_H_4_R)(X)(NHC) (R = H or alkyl, X = Cl, Br, I)] complex can also be used as a catalyst for the polymerization of MMA [69]. The Ni(II) complex or MAO alone does not induce the polymerization, indicating the polymerization must be conducted by both the participation of Ni(II) complexes and MAO. The effect of catalyst on the molecular weight of PMMA is investigated. After a polymerization time of 3 h at 50 °C, the ultrahigh molecular weight PMMA with a *M*_n_ of 1.50 × 10^6^ Da has been successfully prepared in the presence of toluene.

### 2.3. Other Polymerizations

Except for conventional free radical polymerization, atom transfer radical polymerization (ATRP) and coordination polymerization, other polymerizations such as reversible addition-fragmentation chain transfer polymerization (RAFT), anionic polymerization and plasma-initiated polymerization are also used to prepare high molecular weight PMMA. However, reports of RAFT, anionic polymerization and plasma-initiated polymerization for preparing high molecular weight PMMA are much lower than radical polymerization.

Rzayev et al. report a RAFT of MMA by using AIBN as an initiator and cyanoisopropyl dithiobenzoate as a RAFT [6]. The effect of pressure (5 or 9 kbar) on the molecular weight of PMMA is investigated. The PMMA with *M*_n_ of 1.50 × 10^5^ Da and 2.02 × 10^5^ Da are synthesized under 9 kbar and 5 kbar, respectively. The effect of feeding ratio on the molecular weight is also investigated. Moreover, the ultrahigh molecular weight PMMA with a *M*_n_ of 1.25 × 10^6^ Da is synthesized with a molar ratio of [MMA]:[RAFT agent]:[AIBN] = 1.20 × 10^4^:1:0.2.

Kitayama and coworkers report an anionic polymerization of MMA using α-lithioisobutyrate (Li-iPrIB) as initiator in the presence of lithium trimethylsilanolate (Me_3_SiOLi) [70]. The molecular weight of PMMA is significantly influenced by the feeding ratio. Moreover, the PMMA with a high molecular weight (*M*_n_ = 8.25 × 10^5^ Da) and low polydispersity index (PDI = 1.16) has been prepared under a molar ratio of [MMA]:[initiator] = 3000:1.

Since it was discovered by Osada and coworkers [71], the plasma-initiated polymerization has attracted a lot of attention of the chemists. It is easy to synthesize ultrahigh molecular weight PMMA with a *M*_n_ upped to 10^7^ Da through plasma-initiated polymerization [71,72]. The plasma-initiated polymerization has a complicated mechanism, and the most accepted mechanism at present is the radical polymerization mechanism. Plasma-initiated polymerization possesses an outstanding advantage for preparing high molecular weight polymers due to its little number of macromolecular chains. Cheng et al. report that the combination of RAFT and plasma-initiated polymerization can realize the polymerization of MMA [73]. The 2-cyanoprop-2-yl-1-dithionaphthalate (CPDN) is synthesized in their group, and then using as a RAFT reducing agent in the plasma-initiated RAFT polymerization of MMA without the use of any other thermal initiators or photoinitiators. The effect of CPDN concentration on the polymerization of MMA is investigated, finding that the controllability of the polymerization is increased with the increase in CPDN concentration. The polymerization of MMA is also realized in the absence of CPDN, and produces PMMA with an ultrahigh molecular weight of 1.41 × 10^6^ Da.

Nanoparticles have high catalytic activity due to their large surface energy. In our group, the polymerization of MMA with organic halide as initiator in the presence of palladium nanoparticles has been investigated. The ultrahigh molecular weight PMMA with a *M*_n_ of 4.65 × 10^6^ Da and a *M*_w_ of 8.08 × 10^6^ Da is synthesized at 70 °C using 2-bromoisobutyric acid ethyl ester (EBiB) as an initiator in the presence of catalytical amount (10.1 ppm) of Pd NPs [74].

As the development of research, much more polymerization methods are reported and used for the preparation of high molecular weight PMMA [75,76]. Some typical experimental results for the preparation of high molecular weight PMMA are summarized in Table 1.

## 3. Applications of High Molecular Weight PMMA

High molecular weight PMMA and its copolymers are important precursors for the preparation of polymer materials with high light transmittance, excellent electrochemical performance and mechanical properties. PMMA and its copolymers or composites are widely used in electronic equipment, medical technology, and polymer membranes areas. The relationship between molecular weight and physical properties is deeply investigated [77,78]. For instance, the mechanical properties of PMMA are significantly affected by the molecular weight (Figure 7). It shows that the tensile strength, fracture surface energy, shear modulus and Young’s modulus increase with the increase in molecular weight (*M*_υ_) up to 10^6^ Da. Therefore, the synthesized high molecular weight PMMA will further expand the application value of PMMA.

### 3.1. Application of High Molecular Weight PMMA in Medical Field

#### 3.1.1. PMMA Bone Cements

Osteoporotic vertebral compression fracture (OVCF) is one of the most common complications of osteoporosis. At present, percutaneous vertebroplasty (PVP) is widely used for the treatment of OVCF. PMMA bone cement shows an important function in PVP. In a PVP surgery, PMMA bone cement is injected into the injured vertebra of the patient, which can quickly relieve pain, stabilize and strengthen the injured vertebra, and restore the height and angle of the injured vertebra [79,80,81]. However, the strength of ordinary PMMA bone cement is weaker than the bone [82]. Therefore, high molecular weight PMMA can use to prepare higher strength PMMA bone cement. PMMA with viscosity-average molecular weights (*M*_υ_) range from 1.7 × 10^5^ Da to 7.5 × 10^5^ Da are successfully used for the preparation of PMMA bone cements [83], indicating that high molecular weight PMMA might be an excellent raw material for PMMA bone cements.

Inadequate strength at the cement/bone interface is one of the main drawbacks of PMMA bone cement in the current orthopedic surgeries. PMMA cement strength, surface roughness properties and osteo-blast cell growth can be improved by incorporating additives such as MgO, chitosan and hydroxyapatite to PMMA [84,85,86]. Khandaker and coworkers have investigated the fracture toughness (K_IC_) of bone-PMMA with nano MgO particles or micro MgO particles, finding that the K_IC_ of bone-PMMA with nano MgO particles and bone-PMMA with micro MgO particles are much higher than the K_IC_ of bone-PMMA [87].

Lin and coworkers report that the bone/cement interfacial strength can be enhanced by partially degradable PMMA/Mg composite bone cement (PMC) [88]. This reinforcement is accomplished via the increase in the osteo-conductivity of PMMA and the enhancement of the mechanical interlocking between bone tissue and the porous PMMA surface. The effects of Mg composition, particle size and content on the injectability, biocompatibility, mechanical, and degradation properties of PMCs are investigated. They find that the biocompatibility, mechanical and degradation properties of PMCs are influenced by the particle size (75–550 μm), concentration (9–17 wt%) and alloy composition of Mg particles. Moreover, antibacterial capabilities are increased owing to the degradation of Mg, resulting in a decrement in the infection rate.

Roldo and coworkers report a preparation and characterization of antibacterial PMMA composite cement [89]. The chitosan (CS) and methacryloyl chitosan (CSMCC) with concentrations ranging from 10 to 30% *w*/*w* are added to PMMA cement, finding that the mechanical behavior of PMMA cement can be modified by the addition of CS and CSMCC. Bioactive PMMA surfaces at the site of implantation are obtained via the addition of amphiphilic molecule phosphorylated 2-hydroxyethylmethacrylate (HEMA-P) in PMMA bone cement [90]. The addition of HEMA-P shows a positive effect with respect to differentiation and proliferation of the osteoblast-like cell (SaOs-2) without the detrimental changes in other properties. The effects of the addition of soluble calcium and carbonate salts on the properties of PMMA bone cement are investigated. A small amount (1–5%) of soluble salts can enhance the clinical performance of bone cement.

#### 3.1.2. PMMA Denture

PMMA resin is commonly used dental material because of the low cost and lightweight performances. However, the properties such as tenacity, fracture strength of PMMA used for dental material should meet a suitable value. Kusy and coworkers find that the fractions of viscosity-average molecular weight less than 10^5^ make no contribution to the plastic toughening of the material [91]. Huggett and coworkers have investigated the relationship between molecular weight and properties of PMMA denture base, finding that denture base systems with weight-average molecular weights >10^5^ display optimum fracture strength properties [92]. Therefore, the high molecular weight PMMA (*M*_n_ > 10^5^ Da) should be a suitable candidate for dental material.

Both heat- and cold-cured PMMA materials are used for the relining of dentures [93]. Heat-cured PMMA has good bonding strength and wear resistance. However, the roughening of the surface results in a difficulty in cleaning of dental material. Similarly, cold-cured PMMA has poor mechanical properties, leaching of monomers, and associated biocompatibility issues [93,94].

Acrylic (PMMA) teeth are a novel dental material which is manufactured by compression or injection molding techniques. Compared to heat-cured PMMA, acrylic teeth are less brittle owing to the high resilience and flexibility. Distinct from porcelain teeth, acrylic teeth are lightweight and do not cause clicking sounds [95]. However, the strength and adhesion of acrylic teeth are still to be considered. The strength can be improved by the use of additive (nanofillers) or using silanized, feldspar-reinforced PMMA.

Alaa Mohammed and coworkers prepare a new composite material via the mixing of eggshell powder and PMMA resin [96]. The effects of different eggshell powder concentration (1%, 3%, 5% and 7% *w*/*w*) on the property of the material have been studied. The tensile properties and fracture toughness are enhanced by the addition of 7% *w*/*w* of eggshell powder, while the elongation percentage at break and impact strength are decreased compared with other specimens. Eggshell has a poor dispersion ability in PMMA and this may cause a formation of agglomerates in the PMMA matrix. This might be a main reason for the decrement in percentage at break and impact strength. A similar result is obtained when using some other nanofillers as additives [97].

The addition of silica to PMMA has a negative impact on flexural strength. However, the flexural strength can be slightly improved by using silanized feldspar as an additive. Raszewski and coworkers find that properties such as Brinell hardness, elastic modulus, maximal displacement, and flexural strength of PMMA modified with silanized feldspar are obviously improved [97]. In addition, the PMMA modified with silanized feldspar has no adverse effect on Isolde impact resistance compared with the conventional acrylic resin. When using silica filler as the additive, the Brinell hardness and elastic modulus of PMMA are increased. However, this causes a significant decrease in the flexural strength and Isolde impact resistance.

Various types of PMMA materials such as heat-cured PMMA, cold-cured PMMA and light-cured PMMA are used for denture repair [98]. The heat-cured PMMA has a better mechanical property than cold-cured PMMA. However, the heat-cured PMMA has some disadvantages such as time consuming and denture warpage. Compared with heat-cured PMMA and cold-cured PMMA, the light-cured PMMA has some advantages of ease of manipulation, controlled polymerization time, no monomer issues, and better mechanical properties [99,100]. In addition, light-cured PMMA has a better repair strength (40–44 MPa) than heat-cured PMMA (21–34 MPa) and cold-cured PMMA (~13 MPa) [101].

### 3.2. Applications of High Molecular Weight PMMA in Optical and Electricity Area

The PMMA-based polymer nanocomposites have attracted the considerable interest of chemists due to their excellent optical, mechanical and electrical properties [102,103,104].

Polymer field-effect transistors (PFETs) have attracted the significant attention of scientists owing to their potential applications in smart card, displays and sensor [105,106]. The property of a polymer thin film transistor is not only affected by semiconductors, but also by gate insulating film. Therefore, suitable insulating gate dielectric film is very important for the investigation of high-performance polymer field-effect transistors. As the development of donor-acceptor (D-A) copolymers, it has long been known that the molecular weights of polymeric semiconductors play significant roles in enhancing performances of polymer field-effect transistors [107]. PMMA is a commonly used dielectric material owing to its excellent electrical properties.

Mao and coworkers reported that the electron and hole mobilities in polymer field-effect transistors can be enhanced by tailoring the molecular weight of polymeric dielectric [107]. PMMA with different molecular weights are used to investigate the electrical properties of polymer field-effect transistors (Table 2).

It shows that the PFETs based on PMMA (*M*_w_ = 1.2 × 10^5^ Da) exhibit a large electron mobility of 0.30 cm^2^ V^−1^ s^−1^ but show a low hole mobility of 0.01 cm^2^ V^−1^ s^−1^ (Table 2, entry 1). When the molecular weight of PMMA increases to 5.5 × 10^5^ Da, hole mobility of the PFETs is greatly increased to 0.18 cm^2^ V^−1^ s^−1^ and electron mobility is also improved to 0.55 cm^2^ V^−1^ s^−1^ (Table 2, entry 2). Moreover, electron mobility and hole mobility are increased to 0.85, 0.35 cm^2^ V^−1^ s^−1^ respectively when using PMMA with a molecular weight of 1.0 × 10^6^ Da as the dielectric (Table 2, entry 3). In addition, when the molecular weight of PMMA increases from 1.2 × 10^5^ to 1.0 × 10^6^ Da, the trap density (*N*_trap_) for electron traps decrease from 5.46 to 1.38 × 10^11^ cm^−2^ and the trap density (*N*_trap_) for hole traps decrease from 3.64 to 1.10 × 10^11^ cm^−2^. Therefore, using high molecular weight PMMA as dielectric is beneficial for simultaneously enhancing electron and hole mobilities. Consequently, high molecular weight PMMA is an excellent candidate for electrical applications.

Organic field effect transistors (OFETs) have also attracted significant attention owing to their potential applications in electronics [108]. Dinaphtho[2,3-b:2′,3′-f]thieno [3,2-b]thiophene (DNTT) based OFET devices with a bilayer dielectric system comprising of poly (vinyl alcohol) (PVA) and poly (methyl methacrylate) (PMMA) are fabricated by Dhar and coworkers [109]. The influence of molecular weight of PMMA on the property of DNTT based OFET is investigated. They find that high molecular weight PMMA devices are more effective for achieving high photosensitivity and responsivity from the transistors.

Recently, the rare earth luminescent materials have attracted considerable attention owing to their potential applications in optoelectronic devices [110]. However, properties such as processing features, thermal stability and mechanical strength of lanthanide complexes are relatively poor. The addition of polymer is an effective method for the improvement in thermal stability and mechanical strength of lanthanide complexes. Kara and coworkers find that the high molecular weight PMMA (*M*_w_ = 3.50 × 10^5^ Da) is an excellent material for the preparation of Sm/PMMA luminescent composite fiber [111]. Compared to the pure Sm(III) complex, the photostability and temperature stability of Sm/PMMA composite fibers are enhanced due to the modification via PMMA matrix. PMMA provides a rigid environment to prevent the decomposition of the pure Sm(III) complex under high temperature and UV irradiation. The luminescent spectra of Sm/PMMA composite fibers display intense characteristic emissions of the Sm^3+^ ion.

The Co doped ZnO nanoparticle (NP) is used to prepare PMMA (*M*_w_ = 3.50 × 10^5^ Da) and poly (ethyl methacrylate) (PEMA) nanocomposite via casting method [112]. The conductivity of the nanocomposite is increased with the increase in nanofiller contents due to the formation of charge transfer complexes. Furtheemore, the dielectric constancy of the nanocomposite is increased with the increase in temperature. Moreover, the nanocomposite shows excellent thermal and electric properties. Therefore, PEMA/PMMA-Co/ZnO polymer electrolyte is a promising candidate for applications in electrochemical devices.

### 3.3. Applications of High Molecular Weight PMMA in Other Areas

Metal injection molding (MIM) is an important method for the production of small parts with complex shape. The key factor in MIM is the selection of the binder, which should permit the mixing and injection molding of feedstocks with high powder loading. Ideally, the debinding should occur in a short time without causing defects. Bakan et al. report that the complexes consisted of PMMA (*M*_n_ = 1.00 × 10^6^ Da), PEG and stearic acid can be used as a binder for the MIM of 316L stainless steel powder [113]. The viscosity of the system can be reduced by the decrease in the molar ratio of PMMA/PEG, which is beneficial to the molding of feedstocks with high powder loading. However, the strength, stiffness and toughness of the molding are decreased after the binder solidifies. Therefore, the PMMA can harden the solidified molding product.

Membrane gas separation technology is widely used in nitrogen purification, oxygen enrichment, hydrogen recovery from reactor purge gas, and stripping of carbon dioxide from natural gas [114]. Compared to traditional gas separation technologies, the membrane gas separation technology is more competitive. The main subject of membrane gas separation technology is the improvement in the permeability and selectivity of polymer membranes [114,115]. PMMA membrane has a good O_2_/N_2_ selectivity, but it is difficult to operate under a high-pressure condition due to its brittleness [116]. Fu and coworkers report that the high molecular weight PMMA films (*M*_w_ = 9.96 × 10^5^ Da) with different free volumes (FFV) are synthesized in different solvents (dichloromethane, ethyl acetate, tetrahydrofuran, butyl acetate, methyl isobutyl ketone) [114]. This membrane can be operated under a high pressure (30 atm) condition. The permeability coefficients of different gases (He, O_2_, N_2_, CO_2_) in the PMMA membrane are studied, finding that the gas permeability is increased with the increase in FFV. Moreover, they have found that the high molecular weight PMMA membranes possess higher gas permeability than the low molecular weight membranes.

## 4. Summary and Outlook

Compared to low molecular weight PMMA, the high molecular weight PMMA has a better mechanical strength. The application value of PMMA is further expanded due to the synthesis of high molecular weight PMMA. This review summarized the application of conventional free radical polymerization, ATRP, RAFT, plasma-initiated polymerization and ionic polymerization on the preparation of high molecular weight PMMA. A series of linear PMMA, star-shaped PMMA and block polymers with molecular weights between 10^5^–10^7^ Da have been prepared and reported by chemists. The mechanisms of some polymerization systems have also been summarized. Finally, this work also proides a brief description of the application of high molecular weight PMMA in medical, optical and electricity, and other areas. We hope that this work can bring more research insights into this field.

## Figures and Tables

**Figure 1 polymers-14-02632-f001:**
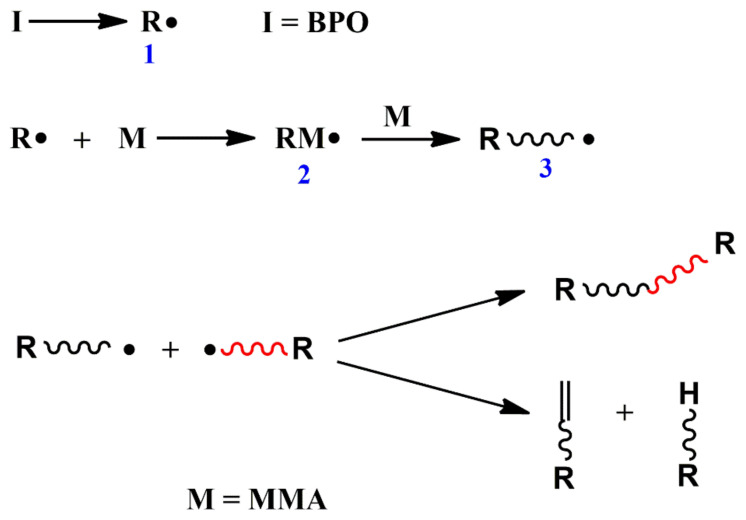
Conventional radical polymerization of MMA initated by BPO.

**Figure 2 polymers-14-02632-f002:**
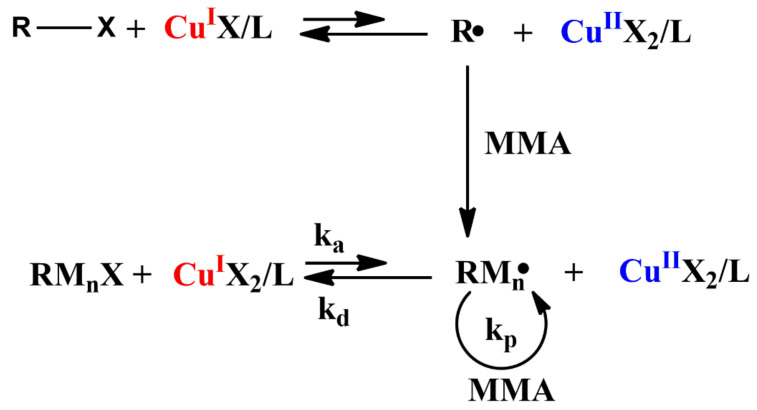
Mechanism of copper mediated normal ATRP of MMA [44].

**Figure 3 polymers-14-02632-f003:**
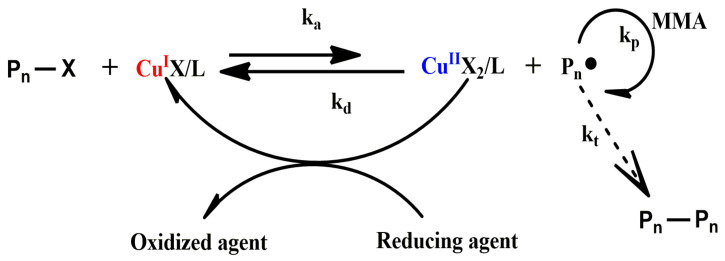
Mechanism of copper mediated ARGET ATRP of MMA (Reprinted with permission from Ref. [49]. Copyright 2007 American Chemical Society).

**Figure 4 polymers-14-02632-f004:**
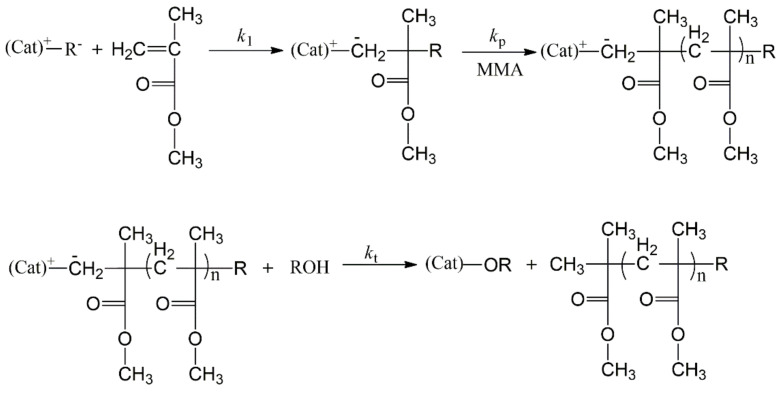
Mechanism of coordination polymerization of MMA.

**Figure 5 polymers-14-02632-f005:**
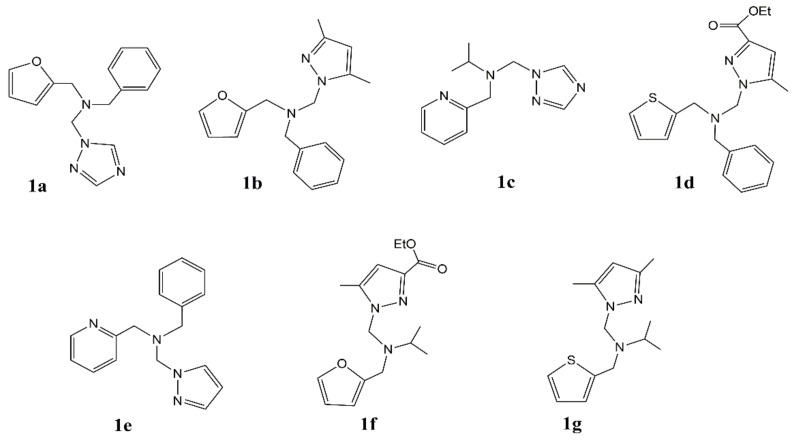
The structure of *N*-tripod ligands (Reprinted with permission from Ref. [63]. Copyright 2007 Elsevier B.V.).

**Figure 6 polymers-14-02632-f006:**
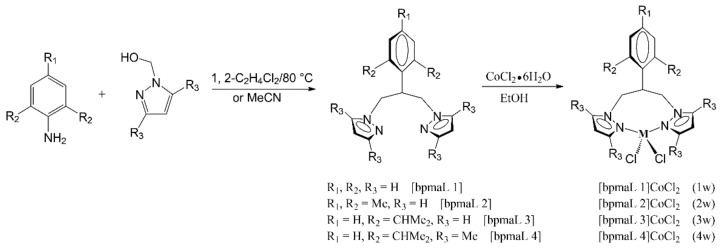
Synthesis of cobalt (II) complex catalyst (Reprinted with permission from Ref. [64]. Copyright 2010 Elsevier B.V.).

**Figure 7 polymers-14-02632-f007:**
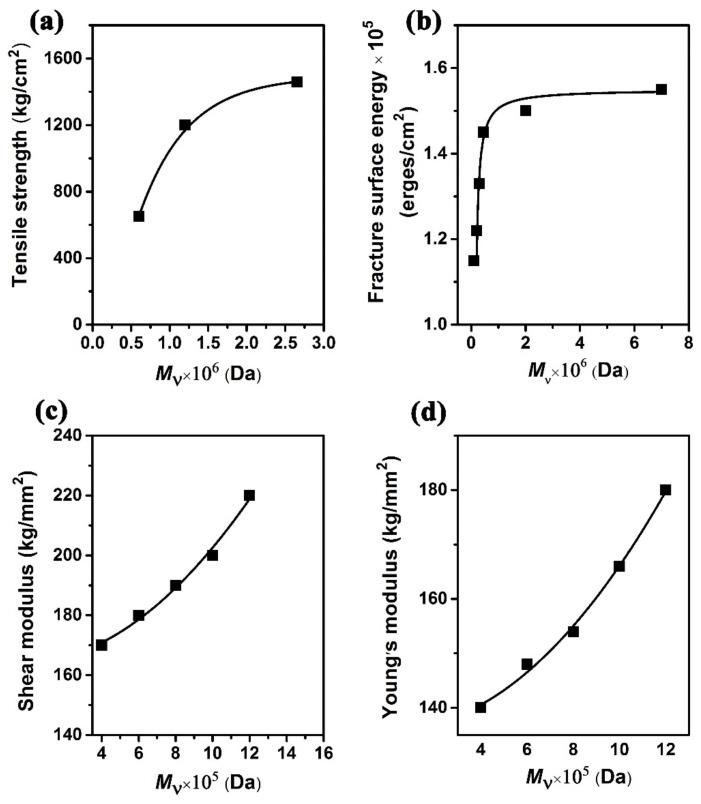
Relationships between mechanical strength ((**a**): Tensile strength; (**b**): Fracture surface energy; (**c**): Shear modulus; (**d**): Young’s modulus) of PMMA and its molecular weight [77,78].

**Table 1 polymers-14-02632-t001:** The results of the polymerization of high molecular weight PMMA.

Entry	Catalyst	Ligand/Additive	Initiator	Temp. (°C)	Time (h)	Conv. (%)	*M*_n_ (g/mol)	*M*_w_/*M*_n_	Refs.
1	/	cyanoisopropyl dithiobenzoate	AIBN	65	7.0	99.0	1.25 × 10^6^	1.03	[6]
2	/	Supercritical CO_2_	AIBN	65	10.0	100.0	1.31 × 10^5^	2.54	[7]
3	/	PVA	ADMVN	25	96.0	83.0	3.61 × 10^6^	2.40	[16]
4	/	Poly(FOA)	AIBN	N/A	4.0	92.0	3.16 × 10^5^	2.09	[17]
5	/	Poly(FOA)	AIBN	65	4.0	88.0	3.65 × 10^5^	2.48	[18]
6	/	P(MMA-co-HEMA)-g-PFPO	AIBN	65	10.0	90.0	3.55 × 10^5^	1.70	[19]
7	/	SDS	APS	70.0	N/A	N/A	5.30 × 10^5^	1.43	[34]
8	/	SDS	AIBN	70	N/A	50.0	1.00 × 10^6^	1.80	[35]
9	/	TEA	Fe/ZnO	N/A	N/A	36.0	2.10 × 10^5^	2.85	[37]
10	/	poly(1-vinyl-3-butylimidazolium ascorbate)	ZnO/Ag	30	9.0	82.0	1.94 × 10^5^	1.38	[42]
11	CuCl	dnNbpy	BMPE	90	2.0	N/A	3.67 × 10^5^	1.20	[43]
12	CuBr	Copper powder	CDB	80	125.0	43.9	1.25 × 10^6^	1.21	[50]
13	CuBr_2_	Tris(2-dimethylaminoethyl)amine	SiO_2_-Br	60	24.0	40.5	1.70 × 10^6^	1.31	[51]
14	CuBr_2_	PMDETA	TBIB	90	4.5	53.0	5.70 × 10^5^	1.36	[52]
15	CuCl	dnNbpy	EBiB	60	24.0	N/A	3.60 × 10^6^	1.24	[53]
16	CuBr_2_	TPMA	EBiB	20	15.0	57.0	1.88 × 10^6^	1.25	[55]
17	Cu(OAc)_2_	MAO	/	30	2.0	31.0	3.39 × 10^5^	N/A	[63]
18	CoCl_2_	*N*,*N*-bis{(1-pyrazolyl)methyl}aniline	/	60	N/A	N/A	1.13 × 10^6^	1.75	[64]
19	Ni(II) complexes	MAO	/	50	3.5	20.0	1.50 × 10^6^	N/A	[69]
20	/	Me_3_SiOLi	Li-iPrIB	−78	1.0	100.0	8.25 × 10^5^	1.16	[70]
21	/	*/*	/	25	16	5.0	1.41 × 10^6^	2.03	[73]
22	Palladium nanoparticles	*/*	EBiB	70	24	82.8	4.65 × 10^6^	1.73	[74]

**Table 2 polymers-14-02632-t002:** Performance parameters of PFET devices using the PMMA dielectric with different molecular weights [107].

Entry	*M*_w_ (Da)	*n*-Channel μ_e,max_ (cm^2^ V^−1^ S^−1^)	*p*-Channel μ_h,max_ (cm^2^ V^−1^ S^−1^)	*n*-Channel *N*_trap_ (×10^11^ cm^−2^)	*p*-Channel *N*_trap_ (×10^11^ cm^−2^)
1	1.2 × 10^5^	0.30	0.01	5.46	3.64
2	5.5 × 10^5^	0.55	0.18	4.67	2.33
3	1.0 × 10^6^	0.85	0.35	1.38	1.10

## Data Availability

Not applicable.

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
