# Peer review of "Development of Synthesis and Application of High Molecular Weight Poly(Methyl Methacrylate)"

_polymers, 2022, doi:10.3390/polym14132632_

Round 1

Reviewer 1 Report

The authors took into account the earlier comments and made the required changes to the manuscript. I recommend this manuscript to the publication in its present form.

Author Response

Response to Reviewer 1 Comments

Point 1: The authors took into account the earlier comments and made the required changes to the manuscript. I recommend this manuscript to the publication in its present form.

Response 1: Thanks a lot for reviewer’s comments.

Reviewer 2 Report

The manuscript has been greatly improved with this revision, as the connection between synthesis conditions and PMMA MW as well as PMMA MW and polymer properties are now described. Nonetheless, there are still a number of minor revisions that are needed, as outlined below:

1.       Line 23 “relatively small density” should be changed to “relatively low density”

2.       Rate coefficients mentioned in lines 69 and 70 should be defined.

3.       Is the polymerization system studied by reference 17 classified as a dispersion polymerization? Since there is a stabilizer added, I don’t think it remains homogeneous.

4.       Line 119 “it brings improvement costs” should be changed to “it brings increased costs”

5.       Lines 158-161: What was the polymerization time required to produced PMMA at 25 C in reference 16?

6.       Line 174-175: I am surprised at the statement that high levels of emulsifier decrease MW of the product. My understanding is that increased emulsifier leads to increased particle number (with reduced size), and thus to increased MW due to radical compartmentalization. Please check this statement and provide a supporting reference if it is not removed.

7.       Line 184: Is the reported MW a number or weight-average value?

8.       Line 189: Is the reported MW a number or weight-average value?

9.       Lines 193-197: I recommend that this new paragraph (describing the need to remove O2 from polymerization systems) be moved to earlier in the section. A better location is following line 92.

10.   Line 291: “A large number of catalysts” should be changed to “a high concentration of catalyst”

11.   Lines 338-339: These two sentences need to be improved (not proper grammatically)

12.   Section 2.2: What are typical reaction times and temperatures for producing high MW PMMA using coordination catalysts? Is it homogeneous or heterogeneous catalysis, and are solvent used? This information (especially compared to reaction times and conditions for radical polymerization) is needed.

13.   Figure 7 plots PMMA properties as a function of Mv.  Is this a viscosity-average MW? Please define this quantity.

14.   Line 503: “As far as we know, there are large numbers of people suffer from osteoporosis.” This is not a proper scientific sentence. It can be completely removed from the manuscript.

15.   I recommend that Table 2 be removed from the paper. It is not needed, as it is sufficient to state in the text that PMMA bone cements have MWs between 1.7 and 7.5 x 10^5 Da. In addition, it appears as these values are also viscosity-average MWs (Mv). This should be stated.

16.   Lines 548-554: The properties of dental material are discussed. However, it is not clear if the MW values mentioned are number- or weight-average values.

17.   Line 608-609: The Mw values should not be listed here, as they are contained in the table.

18.   Line 685-686: The statement makes it sound as if PMMA was prepared as part of this study. It can be combined with the previous sentence “… for the preparation linear, star-shaped or block PMMA structures with high molecular weights between 10^5 and 10^7 Da.”

Author Response

Response to Reviewer 2 Comments

Point 1: Line 23 “relatively small density” should be changed to “relatively low density”

Response 1: Thanks a lot for reviewer’s suggestion. We have changed the “relatively small density”to “relatively low density” in the revised manuscript.

Point 2: Rate coefficients mentioned in lines 69 and 70 should be defined.

Response 2: As the reviewer suggested. We have defined the rate coefficients in line 69-71 in the revised manuscript.

Point 3: Is the polymerization system studied by reference 17 classified as a dispersion polymerization? Since there is a stabilizer added, I don’t think it remains homogeneous.

Response 3: Thanks a lot. We perused this article titled “Dispersion polymerization of methyl methacrylate stabilized with poly(1,1-dihydroperfluorooctyl acrylate) in supercritical carbon dioxide” again (Macromolecules 1995, 28, 8159-8166). Hsiao and coworkers believe that this polymerization can be classified as dispersion polymerization.

Point 4: Line 119 “it brings improvement costs” should be changed to “it brings increased costs”

Response 4: Thanks a lot for reviewer’s suggestion. We have revised this in line 126 in the revised manuscript.

Point 5: Lines 158-161: What was the polymerization time required to produced PMMA at 25 C in reference 16?

Response 5: Thanks a lot. It needs 96 h for preparing high molecular weight PMMA at 25 °C. We have described it in line 169 in the revised manuscript.

Point 6: Line 174-175: I am surprised at the statement that high levels of emulsifier decrease MW of the product. My understanding is that increased emulsifier leads to increased particle number (with reduced size), and thus to increased MW due to radical compartmentalization. Please check this statement and provide a supporting reference if it is not removed.

Response 6: Thank you very much. There is no doubt that the reviewers are right. As the reviewer stated, increasing the emulsifier will cause an increment of molecular weight. We originally wanted to express that increasing the concentration of emulsifier will cause a decrement of the property of the synthesized PMMA. Unfortunately, we made a mistake. Therefore, we have removed this sentence in the revised manuscript.

Point 7: Line 184: Is the reported MW a number or weight-average value?

Response 7: Thanks a lot. It is the weight-average molecular weight. We have revised it in line 191 in the revised manuscript.

Point 8: Line 189: Is the reported MW a number or weight-average value?

Response 8: Thanks a lot. It is the number-average molecular weight. We have revised it in line 196 in the revised manuscript.

Point 9: Lines 193-197: I recommend that this new paragraph (describing the need to remove O2 from polymerization systems) be moved to earlier in the section. A better location is following line 92.

Response 9: Thanks a lot for reviewer’s suggestion. We have moved this paragraph to the location suggested by the reviewer.

Point 10: Line 291: “A large number of catalysts” should be changed to “a high concentration of catalyst”

Response 10: Thanks a lot for reviewer’s suggestion. We have revised it in line 293 in the revised manuscript.

Point 11: Lines 338-339: These two sentences need to be improved (not proper grammatically)

Response 11: Thanks a lot. We have modified these two sentences in line 340-341 in the revised manuscript.

Point 12: Section 2.2: What are typical reaction times and temperatures for producing high MW PMMA using coordination catalysts? Is it homogeneous or heterogeneous catalysis, and are solvent used? This information (especially compared to reaction times and conditions for radical polymerization) is needed.

Response 12: Thanks a lot for reviewer’s suggestion. We have added more description on the coordination polymerization in line 366-384 in the revised manuscript. In addition, some conditions such as the polymerization time and the polymerization temperature of the polymerization described in section 2.2 have been provided.

Point 13: Figure 7 plots PMMA properties as a function of Mv. Is this a viscosity-average MW? Please define this quantity.

Response 13: Thanks a lot. As the reviewer implied, the Mυ is the viscosity-average molecular weight. We have defined it in line 173 in the revised manuscript.

Point 14: Line 503: “As far as we know, there are large numbers of people suffer from osteoporosis.” This is not a proper scientific sentence. It can be completely removed from the manuscript.

Response 14: Thanks a lot for reviewer’s suggestion. This sentence has been removed.

Point 15: I recommend that Table 2 be removed from the paper. It is not needed, as it is sufficient to state in the text that PMMA bone cements have MWs between 1.7 and 7.5 x 10^5 Da. In addition, it appears as these values are also viscosity-average MWs (Mv). This should be stated.

Response 15: Thanks a lot for reviewer’s suggestion. Table 2 has been removed from the paper. As the reviewer implied, these values are viscosity-average molecular weight. We have revised this sentence in line 538-540 in the revised manuscript.

Point 16: Lines 548-554: The properties of dental material are discussed. However, it is not clear if the MW values mentioned are number- or weight-average values.

Response 16: Thanks a lot. We have revised it in line 575 and line 578 in the revised manuscript.

Point 17: Line 608-609: The Mw values should not be listed here, as they are contained in the table.

Response 17: Thank you very much. As the reviewer suggested, the Mw values have been removed.

Point 18: Line 685-686: The statement makes it sound as if PMMA was prepared as part of this study. It can be combined with the previous sentence “… for the preparation linear, star-shaped or block PMMA structures with high molecular weights between 10^5 and 10^7 Da.”

Response 18: Thanks a lot. These sentences have been revised in line 707-711 in the revised manuscript.

This manuscript is a resubmission of an earlier submission. The following is a list of the peer review reports and author responses from that submission.

Round 1

Reviewer 1 Report

A review manuscript by Yan et al. is devoted to polymethyl methacrylate, a polymer that has been studied and widely used in industry. However, to highlight the work, the authors focused on high molecular polymethyl methacrylate (PMMA), methods for its synthesis and applications in various fields. The manuscript includes a review of the main methods for obtaining high molecular PMMA (free radical and coordination polymerization, ATRP, RAFT), and also lists the application areas of PMMA, in which the emphasis is on applications in the field of medicine. In addition, the authors included the results of their research in the text.

The submitted manuscript covers the stated topics quite well and in detail. It is well written and logically structured. However, there are also some comments from me.

1. On page 2, lines 62-67 and Fig. 1 describes the mechanism of free radical polymerization (FRP) and shows it graphical interpretation of this process. However, when it comes to more complex mechanisms, such as ATRP, RAFT, etc., there is no such detailed description. Therefore, the authors should remove the well-known and redundant information about the mechanism of FRP, because this is a scientific work addressed to specialists in the field of polymers.

2. Section 2.1.2 deals with ATRP. The introduction to it contains a number of references to the works of various authors. Further in the text, specific achievements and successes in the use of the ATRP mechanism in the synthesis of PMMA are mentioned, including the publication of the discoverer of this mechanism, Krzysztof Matyjaszewski. They are located at the very end of the section. In my opinion, this is extremely illogical and needs to be changed. The works of the founder should be listed among the first ones considered in the section, and should also be mentioned in the preamble. In addition, figure 2 needs to be changed: its representation is very similar to that used by Matyjaszewski in his work. Either modify it so that there is no hint of plagiarism, or include a link to Matyjaszewski's work in the caption.

3. Section 3.1 is devoted to the application of high molecular weight PMMA in various fields. However, practically nowhere in the text is the molecular weight of the used polymers or the advantages of high molecular PMMA over low molecular one in this area of use.

Based on the foregoing, I conclude that the manuscript of Yan and co-authors can be accepted for publication after these shortcomings are eliminated.

Reviewer 2 Report

This article fails to provide any evidence linking PMMA MW to material performance. For example, the focus of Section 3.1 is on the addition of additives to PMMA to improve properties of bone cement and dental material. But as far as the reader can tell, this work might be done with low MW PMMA. The same is true in Section 3.2 (optical and electrical applications): the section also does not contain a single example of why high MW PMMA is required to achieve the desired properties. Thus, the stated purpose of the manuscript is not met. To convince the reader that these specialty applications are only possible with high MW material, the authors need to: (i) provide data that link important PMMA physical properties to polymer MW, and (ii) demonstrate that PMMA with lower MW cannot be used for the described applications. Without this information, the submission should be rejected.

In addition, Section 2 (“Synthesis of High Molecular Weight PMMA”) of the manuscript is not useful. It is simply a list of studies that have produced high MW polymer, without any analysis provided. Why were these specific reaction conditions required to make material with high MW? For example:

-          in conventional radical polymerization, it is necessary to operate at a very low radical concentration to reduce the rate of radical-radical termination. While this can be achieved, the tradeoff is that much longer reaction times are required in solution or suspension systems.

-          For ATRP (or other controlled radical polymerization techniques), it is necessary to set a high monomer to initiator ratio. However, this also leads to slowed reaction rates, and the conditions must be chosen to reduce the importance of chain transfer or other side reactions.

This context is needed in addition to providing the literature examples.

Other comments:

1.      I suggest that the authors change their tense from past to present throughout the manuscript. A sentence like “Free radical polymerization was one of the most important technologies …” implies that it is no longer used commercially, which is not the case.

2.       The authors need to define what they mean by high molecular weight. For example, is it PMMA with number average MW > 10^5 Dalton? Is it weight average MW > 10^6 Da? And what physical properties of the polymers improve when this threshold is achieved?

3.       Production of PMMA in supercritical CO2 is a heterogeneous (dispersion) process. Thus, it is incorrect to refer to CO2 as a “solvent” (line 82). It is better to use the term diluent or reaction medium.

4.       Line 122-124: The discussion of poly(acrylonitrile) should be removed.

5.       A few sentences are needed to explain the ATRP mechanism (role of the mediating agent, relative rates of activation and deactivation, all chains initiated at the start of reaction, linear growth of MW with conversion, low dispersity, …) shown as Figure 2. Both “normal ATRP” and ARGET ATRP are mentioned in the section, but the differences (role of reducing agent in ARGET ATRP) are not well-described.

6.       Is control of tacticity, only achievable by coordination polymerization and not by radical polymerization, important? This can also greatly change the properties of PMMA important for the applications described in Section 3.

7.       Table 1 includes a column of MW data with “Mn or Mw”. It is not acceptable to group these together: the values can’t be directly compared, as there can be a large difference between Mn and Mw (especially for conventional radical polymerization).